# Comparison of Preference for Chemicals Associated with Fruit Fermentation between *Drosophila melanogaster* and *Drosophila suzukii* and between Virgin and Mated *D. melanogaster*

**DOI:** 10.3390/insects14040382

**Published:** 2023-04-14

**Authors:** Hyemin Kim, YeongHo Kim, Gwang Hyun Roh, Young Ho Kim

**Affiliations:** 1Department of Ecological Science, Kyungpook National University, Sangju-si 37224, Gyeongsangbuk-do, Republic of Korea; 2Department of Plant Medicine and Institute of Agriculture & Life Sciences, Gyeongsang National University, Jinju-si 52828, Gyeongsangnam-do, Republic of Korea; 3Research Institute of Invertebrate Vector, Kyungpook National University, Sangju-si 37224, Gyeongsangbuk-do, Republic of Korea

**Keywords:** *Drosophila melanogaster*, *Drosophila suzukii*, chemical preference, electroantennogram, virgin, mated female

## Abstract

**Simple Summary:**

Different species of *Drosophila* show evolutionary specializations in host preference, habitat choice, and morphology. Therefore, the selection of different species belonging to the *Drosophila* genus is advantageous for understanding evolutionary adaptations to certain environments. In particular, two taxonomically close species, *Drosophila melanogaster* and *Drosophila suzukii*, are known to have distinct habitats; *D. melanogaster* is mostly found near overripe, decaying, abandoned, and fermented fruits, whereas *D. suzukii* is attracted to fresh fruits. The infection of microorganisms promotes fruit fermentation accompanied by the production of large amounts of chemicals. Therefore, chemical concentrations are typically higher in fermented fruits than in fresh fruits. Considering the distinct habitats of the two flies, *D. melanogaster* and *D. suzukii* are thought to be more attracted to high and low concentrations of chemicals, respectively. In this study, Y-tube olfactometer and electroantennogram assays revealed that *D. melanogaster* had a higher preference for relatively high concentrations of 2-phenylethanol, ethanol, and acetic acid than that of *D. suzukii*. In the comparison between virgin and mated females of *D. melanogaster*, mated flies were more attracted to high concentrations of chemicals than virgin flies. These results suggest that high concentrations of chemicals are an important attraction factor for *D. melanogaster* seeking appropriate sites for oviposition.

**Abstract:**

Two taxonomically similar *Drosophila* species, *Drosophila melanogaster* and *Drosophila suzukii*, are known to have distinct habitats: *D. melanogaster* is mostly found near overripe and fermented fruits, whereas *D. suzukii* is attracted to fresh fruits. Since chemical concentrations are typically higher in overripe and fermented fruits than in fresh fruits, *D. melanogaster* is hypothesized to be attracted to higher concentrations of volatiles than *D. suzukii*. Therefore, the chemical preferences of the two flies were compared via Y-tube olfactometer assays and electroantennogram (EAG) experiments using various concentrations of 2-phenylethanol, ethanol, and acetic acid. *D. melanogaster* exhibited a higher preference for high concentrations of all the chemicals than that of *D. suzukii*. In particular, since acetic acid is mostly produced at the late stage of fruit fermentation, the EAG signal distance to acetic acid between the two flies was higher than those to 2-phenylethanol and ethanol. This supports the hypothesis that *D. melanogaster* prefers fermented fruits compared to *D. suzukii*. When comparing virgin and mated female *D. melanogaster*, mated females showed a higher preference for high concentrations of chemicals than that of virgin females. In conclusion, high concentrations of volatiles are important attraction factors for mated females seeking appropriate sites for oviposition.

## 1. Introduction

Species belonging to the *Drosophila* genus show evolutionary specialization in host preference, habitat choice, and morphology [1]. In particular, two taxonomically similar fruit fly species, *Drosophila melanogaster* Meigen and *Drosophila suzukii* Matsumura, belonging to the same family Drosophilidae and melanogaster group [2] have distinct habitats. Like most fruit fly species, *D. melanogaster* shows a preference for volatile chemicals associated with fermentation, such as ethanol, acetic acid, ethyl acetate, and acetaldehyde [3]. Since these flies are predominantly found near fruits that are abandoned and undergo fermentation after harvest, *D. melanogaster* is not considered an economically serious pest [3]. Unlike the majority of the members of *Drosophila*, *D. suzukii* is evolutionarily specialized to possess a preference for undamaged, freshly ripening fruits [4]. Since female *D. suzukii* damages fresh fruits by penetrating the fruit surface for oviposition via its serrated ovipositor, it is widely considered a serious pest in North America and Europe [5].

During the overripening and fermentation processes, various volatile chemicals such as 2-phenylethanol, ethanol, acetic acid, and methanol are produced [3,6,7]. Microorganisms such as bacteria and yeast play an important role in inducing ripening and fermentation in fruits, thereby resulting in a higher concentration and quantity of the volatile compounds in overripe and fermented fruits than those in freshly ripening fruits [8,9,10,11]. In particular, yeast mainly produces large amounts of 2-phenylethanol, ethanol, and acetic acid via fermentation [9,10].

The concentration of volatile compounds differs depending on the fruit ripening stages [12]. Therefore, based on the distinct habitats of the two *Drosophila* species, *D. melanogaster,* which is constantly exposed to these chemical stressors [3], is thought to have evolutionarily adapted to be tolerant to the high concentration of chemicals, including 2-phenylethanol, ethanol, and acetic acid, produced in its rotting-fruit dietary niche [13]. In contrast, *D. suzukii*, which inhabits fresh fruit, is exposed to relatively low chemical concentrations. Therefore, *D. melanogaster* and *D. suzukii* were hypothesized to be tolerant and susceptible to these chemicals, respectively. We recently demonstrated that *D. melanogaster* showed higher survival rates than *D. suzukii* when they were treated with high concentration of 2-phenylethanol, ethanol, and acetic acid, although they were reared in typical artificial diets [14,15]. Previous studies have shown that high expression levels or enzymatic activities of alcohol dehydrogenase, acetyl-CoA synthase, phospholipase D, ornithine aminotransferase (*OAT*), and glutathione-S-transferase (*GST*) contribute to chemical tolerance in *D. melanogaster* [13,16,17,18,19]. In particular, in the comparison between the two flies, higher expression levels of *OAT* and *GST* are found in *D. melanogaster* than those in *D. suzukii.* This supported the fact that *D. melanogaster* has adapted to tolerate nitrogen waste, including ammonia and urea, which are produced and accumulated via microbial metabolism in the fruit fermentation process [18,19]. Our recent study further showed that different induction levels of antimicrobial peptides upon chemical exposure in *D. melanogaster* and *D. suzukii* contribute to the different chemical tolerances of the two flies [15].

Regarding the different habitats of *D. melanogaster* and *D. suzukii*, these two flies are thought to possess distinct host preferences in addition to chemical tolerance. When oviposition behavior has previously been compared among *D. suzukii*, *D. melanogaster*, and other closely related species using rotten and ripe strawberries, the two flies have been found to demonstrate contrasting behavior: *D. suzukii* lays almost all its eggs on ripe fruit, whereas *D. melanogaster* females select rotten fruit almost exclusively [20,21]. Moreover, in a comparison of the oviposition preference for ethanol and acetic acid between the two flies, although the two flies do not show different oviposition preferences for ethanol, *D. melanogaster* shows a high preference for acetic acid at each concentration tested, whereas *D. suzukii* avoids high concentrations of acetic acid, which is mostly produced in the late stage of fruit maturation or fermentation [21]. Therefore, the two flies may have different preferences for different types and concentrations of chemicals produced, depending on the stage of fruit fermentation.

Notably, in *D. suzukii*, previous studies have reported contrasting results for host preference. Female *D. suzukii* have been reported to be more attracted to fresh strawberries for oviposition than overripe strawberries [20]; however, in another study, they showed preference for overripe strawberries over red strawberries [22]. This discrepancy in host preference in *D. suzukii* females has been suggested to be due to the fact that mate- or food-seeking flies are more attracted to overripe fruit volatiles, whereas ovipositing female flies exhibit a preference for fresh fruit volatiles [23]. Therefore, the chemical preference can be altered depending on the status of females.

Considering that *D. melanogaster* and *D. suzukii* have different host preferences, we compared the preferences of the two flies for different concentrations of volatile chemicals emitted from overripe and fermenting fruits, including 2-phenylethanol, ethanol, and acetic acid, using Y-tube olfactometer and electroantennogram (EAG) assays. In addition, to investigate the chemical preference properties depending on the physiological status of the females, we performed Y-tube and EAG experiments using virgin and mated female *D. melanogaster* in the present study.

## 2. Materials and Methods

### 2.1. Insects

The wild-type strain Canton-S *D. melanogaster* used in this study was obtained from the Bloomington Drosophila Stock Center (Indiana University, Bloomington, IN, USA). *D. suzukii* was a gift from Gyeongsang National University (Jinju, Gyeongnam, Republic of Korea). The flies were maintained at 25 ± 2 °C with 50–70% relative humidity and a light: dark cycle of 16:8 h as described previously [14,15,24]. Larvae were reared in a Drosophila vial containing standard medium (Bloomington Stock Center Recipe). Emerging adults were transferred to a BugDorm (24 × 24 × 24 cm) (BioQuip, Rancho Dominguez, CA, USA) containing standard media for oviposition. 

To obtain mated females of *D. melanogaster* and *D. suzukii*, approximately 150 newly eclosed flies including similar numbers of males and females were placed in a BugDorm for 3 d and allowed to mate. Virgin females of *D. melanogaster* were obtained within 2 h after emergence (sexually immature) after observation of the meconium under a stereomicroscope [25]. To perform the Y-tube attraction assay and EAG experiments, 3- to 5-day-old mated females of *D. melanogaster* and *D. suzukii* were used to compare chemical preferences between the two fly species. As females of *D. melanogaster* are unreceptive for mating until about 24 h after eclosion [26], <1-day-old virgins were used to compare chemical preferences between virgin and mated flies.

### 2.2. Y-Tube Attraction Test

Olfactory responses were compared between mated females of *D. melanogaster* and *D. suzukii* and between the reproductive status (virgin vs. mated) of female *D. melanogaster* using a glass Y-tube olfactometer (stem, 20 cm; arm length, 15 cm; internal diameter, 2.5 cm; arm angle, 45°). Each arm of the Y-tube was connected to a glass jar (250 mL) containing deionized water. Activated charcoal-filtered and humidified clean air was then pumped through the Y-tube. The flow rate of the pump was set to 0.5 L/min.

A two-choice assay was used to evaluate the preferences of the flies for various concentrations of three chemicals (2-phenylethanol, ethanol, and acetic acid) at 25 ± 2 °C. Three chemicals were diluted in deionized water to prepare the stock solutions (1% 2-phenylethanol, 10% ethanol, and 10% acetic acid). The chemical stock solutions were then serially diluted to various concentrations (0.01–0.1% for 2-phenylethanol; 1–10% for ethanol; 0.1–1% for acetic acid). For the two-choice test, 2 mL of each chemical solution was applied to the cotton roll placed in each arm of Y-tube at various concentrations (%): 0.01 vs. 0.1, 0.02 vs. 0.09, 0.03 vs. 0.08, 0.04 vs. 0.07, and 0.05 vs. 0.06 for 2-phenylethanol; 1 vs. 10, 2 vs. 9, 3 vs. 8, 4 vs. 7, and 5 vs. 6 for ethanol; 0.1 vs. 1, 0.2 vs. 0.9, 0.3 vs. 0.8, 0.4 vs. 0.7, and 0.5 vs. 0.6 for acetic acid. Twenty flies were introduced into the entrance of the olfactometer, and the number of flies that entered the side arms was counted after 45 min. Flies that remained in the main stem for 45 min were recorded as “no choice”. All experiments were conducted in triplicate. The positions of the arms were switched after each replication to avoid bias. The chemical solutions were replaced with freshly prepared solutions after each experiment. The olfactometer was rinsed with water, 70% ethanol, autoclaved, and heated overnight at 60 °C after three replications of each pair of chemical concentrations.

### 2.3. EAG Experiments

The EAG responses were compared between *D. suzukii* and *D. melanogaster* and between virgin and mated female *D. melanogaster* to three chemicals (2-phenylethanol, ethanol, and acetic acid). For the response test, serial dilutions (0.01, 0.02, 0.04, 0.06, 0.08, and 0.1% for 2-phenylethanol; 1, 2, 4, 6, 8, and 10% for ethanol; 0.1, 0.2, 0.4, 0.6, 0.8, and 1% for acetic acid) of each chemical were prepared in deionized water using the same method as for the Y-tube attraction test. Each antenna of the flies was prepared by excising the thorax. For odor presentation, a piece (0.3 × 2.5 cm) of filter paper (Advantec No. 2; Advantec, Tokyo, Japan) loaded with 10 µL of each test solution concentration was inserted into a glass Pasteur pipette (150 mm in length; Hilgenberg, Malsfeld, Germany). A glass capillary (1.1 mm I.D.; Paul Marienfeld, Lauda-Königshofen, Germany) filled with electro-conductive gel (Spectra 360; Parker Laboratory Inc., Orange, NJ, USA) was connected to the excised thorax as a reference electrode. Another glass capillary containing electroconductive gel, which served as the recording electrode, was connected to the tip of the antennal segment. The EAG signal was recorded using a PC-based signal processing system (IDAC-2; Syntech, Hilversum, The Netherlands) and GC-EAD 2000 software (Syntech). Next, 10 mL of purified, charcoal-filtered air was introduced through the Pasteur pipette cartridge for 0.5 s into a stainless-steel tube via a continuous humidified main air stream (600 mL/min) using an electrically controlled airflow controller (CS-05, Syntech). Successive stimulations were performed after 30 s intervals. Stimulation was performed at low-to-high doses. Each odor-stimulus cartridge was used once. Each antenna was exposed to all concentrations from a single chemical starting with the lowest and ending with the highest concentration. A single antenna was used for each replicate, and the antenna was changed after one replicate was completed. Three replicates were performed for each chemical.

### 2.4. Statistical Analysis

Data were analyzed using SPSS for Windows version 25.0 (IBM, Armonk, NY, USA). Trends in EAG response to different concentrations of chemicals between *D. suzukii* and *D. melanogaster* and between virgin and mated *D. melanogaster* were compared using GLM-repeated measures of analysis of variance (ANOVA). The attraction rate and EAG response at a given concentration of chemical were compared using the Student *t*-test for comparisons between the two fly species and between virgin and mated *D. melanogaster*. In addition, Student’s *t*-test was used to compare the average EAG response values between the two flies and between virgin and mated *D. melanogaster*. The ratio of EAG responses (*D. melanogaster/D. suzukii* and mated/virgin *D. melanogaster*) was statistically analyzed using a one-way ANOVA. All data were presented as the mean ± standard error.

## 3. Results

### 3.1. Comparison of Preference for Chemicals at Various Concentrations between D. melanogaster and D. suzukii

In the comparison of preferences for different concentration of chemicals between two flies, *D. melanogaster* and *D. suzukii* exhibited significantly difference preferences to each chemical and concentration range tested using a Y-tube assay (Figure 1 and Appendix A). Among various concentration of 2-phenylethanol (0.01–0.1%), *D. melanogaster* preferred a higher concentration (0.06–0.1%) than *D. suzukii* did, whereas *D. suzukii* showed a higher preference for 0.03% 2-phenylethanol (*f* = 0.062, *df* = 1, *p* < 0.05) (Figure 1a). In the comparison of preferences for 1–10% ethanol, *D. melanogaster* showed a higher preference for 8–10% ethanol than that of *D. suzukii*; in contrast, *D. suzukii* showed a higher preference for ethanol at concentrations of 1 and 3% than that of *D. melanogaster* (Figure 1b). Similarly, *D. melanogaster* showed a higher preference for high concentrations of acetic acid (0.5 and 0.8–1%), whereas, although it was not significantly different between the two flies, *D. suzukii* showed a slightly higher preference for low concentrations of acetic acid (0.1–0.4%) (Figure 1c). When we examined the linear relationship across preference for various concentrations of the three chemicals between the two flies using simple regressions, the slopes of the linear regression equation for *D. melanogaster* were 0.0551, 0.0465, and 0.039 in 2-phenylethanol, ethanol, and acetic acid treatment, respectively, whereas those for *D. suzukii* were −0.0185, −0.0228, and −0.0321, respectively. This suggests that *D. melanogaster* preferred high concentrations of the chemicals, whereas *D. suzukii* showed a slightly higher preference for low concentrations of the chemicals produced during fruit fermentation (Figure 1).

In addition to the Y-tube attraction assay, the EAG responses of *D. melanogaster* and *D. suzukii* to various concentrations of 2-phenylethanol, ethanol, and acetic acid were compared (Figure 2 and Figure 3). The antenna of *D. melanogaster* showed a significantly greater EAG response to 2-phenylethanol at 0.1% (*f* = 8.163, *df* = 1, *p* = 0.046) and acetic acid at 0.1% (*f* = 7.812, *df* = 1, *p* = 0.049), 0.2% (*f* = 16.962, *df* = 1, *p* = 0.015), 0.6% (*f* = 8.124, *df* = 1, *p* = 0.046) 0.8% (*f* = 18.917, *df* = 1, *p* = 0.012), and 1.0% (*f* = 2.915, *df* = 1, *p* = 0.009) than that of *D. suzukii* antenna (Figure 1a,c). Although the EAG responses to ethanol between the two fly species were not significantly different at all concentrations, *D. melanogaster* antenna exhibited higher EAG responses than those of *D. suzukii* (Figure 2b). When the overall EAG responses across different concentrations of each chemical between the two flies were compared, although no significant differences were obtained, *D. melanogaster* consistently demonstrated greater EAG responses to all concentrations of chemicals than those of *D. suzukii* (Figure 2). In particular, as judged by the slopes of the linear regression equations for the two flies, the antenna of *D. melanogaster* displayed concentration-dependent EAG responses to the three chemicals. *D. suzukii* showed concentration-dependent EAG responses to ethanol and acetic acid, but its slope to 2-phenylethanol was relatively flat compared to the other two chemicals (Figure 2). The slopes for *D. melanogaster* were 3.8- and 3.1-fold higher than those for *D. suzukii* in the 2-phenylethanol and acetic acid treatments, respectively (Figure 2a,c). In contrast, *D. suzukii* had a 1.2-fold greater slope for ethanol than *D. melanogaster* (Figure 2b). Based on the values of the slopes, a comparison of the EAG responses to various concentrations of each chemical indicated that *D. melanogaster* was generally more responsive to high concentrations of 2-phenylethanol and acetic acid than *D. suzukii*, whereas the difference in ethanol preference between the two flies was not significant (Figure 2).

When the average values of the EAG response signals were calculated for every concentration used in this study, all three compounds elicited significantly greater EAG responses in the antenna of *D. melanogaster* than in the antenna of *D. suzukii* (*f* = 1.721, *df* = 34, *p* = 0.002 for 2-phenylethanol; *f* = 0.000, *df* = 34, *p* = 0.015 for ethanol; *f* = 7.406, *df* = 34, *p* < 0.001 for acetic acid) (Figure 3a). This further suggested that *D. melanogaster* had a higher antenna sensitivity toward these chemicals than that of *D. suzukii*. In addition, when the ratio of the EAG responses (*D. melanogaster* response/*D. suzukii* response) was calculated, *D. melanogaster* exhibited approximately 1.45-, 1.47-, and 2.44-fold higher EAG responses to 2-phenylethanol, ethanol, and acetic acid, respectively, than those of *D. suzukii* (Figure 3b). Although the EAG responses were not significantly different between the three tested compounds (*f* = 2.104, *df* = 2, *p* = 0.203), the response distance between the two flies to acetic acid was greater than those for 2-phenylethanol and ethanol (Figure 3b). These findings suggest that *D. melanogaster* has a higher antenna sensitivity to acetic acid than for the other two chemicals compared with that of *D. suzukii*.

### 3.2. Comparison of Preference for Chemicals at Various Concentrations between Virgin and Mated D. melanogaster

Since a comparison between the two species revealed that *D. melanogaster* showed higher attraction and EAG responses to the three chemicals than those of *D. suzukii*, we next compared the chemical preference between virgin and mated female *D. melanogaster* (Figure 4 and Appendix A). In general, mated *D. melanogaster* showed a higher preference for high concentrations of all three chemicals than that of virgin flies in the Y-tube olfactometer assay (Figure 4). According to linear regression analysis, the slopes of mated *D. melanogaster* were 0.0551, 0.0465, and 0.039 and those of virgin *D. melanogaster* were 0.02, −0.048, and −0.0122 for 2-phenylethanol, ethanol, and acetic acid, respectively (Figure 4 and Appendix A). This indicated that mated *D. melanogaster* showed higher preference for higher concentrations of chemicals than those of virgin females. In the 2-phenylethanol choice test, virgin and mated *D. melanogaster* showed similar attraction trends; however, significantly more mated females were attracted to 0.06 and 0.1% 2-phenylethanol than were virgin females (*f* = 12.5, *df* = 1, *p* = 0.024 and *f* = 35.579, *df* = 1, *p* = 0.004, respectively; Figure 4a). In contrast to the 2-phenylethanol trends, the two females exhibited contrasting preference for ethanol and acetic acid: virgin flies showed higher preference for low concentrations of ethanol and acetic acid, whereas mated *D. melanogaster* showed higher preference for higher concentrations of chemicals (Figure 4b,c). Particularly, virgin flies showed higher preference for 2% and 3% ethanol (*f* = 17.818, *df* = 1, *p* = 0.013 and *f* = 16.568, *df* = 1, *p* = 0.015, respectively), while mated flies showed higher preference for 7% (*f* = 12.291, *df* = 1, *p* = 0.025), 9% (*f* = 13.520, *df* = 1, *p* = 0.021), and 10% ethanol (*f* = 14.535, *df* = 1, *p* = 0.019) (Figure 4b). Similarly, although it was not significantly different, more mated females were found near high concentrations of acetic acid (0.6–1%), whereas virgin flies were more attracted to the lowest concentration of acetic acid (0.1%) (*f* = 8.643, *df* = 1, *p* = 0.042) (Figure 4c). Taken together, our Y-tube attraction assay revealed that mated *D. melanogaster* possess a preference for high concentrations of chemicals compared to virgin females.

Similar to the Y-tube olfactory assay, the EAG responses of virgin and mated *D. melanogaster* revealed that mated females had higher antenna responses to high chemical concentrations than those of virgin females (Figure 5). Among the three chemicals, ethanol and acetic acid elicited significantly greater EAG responses in the antenna of mated *D. melanogaster* than in the antenna of virgin *D. melanogaster* at all concentrations, except 1% ethanol (*p* < 0.05; Figure 5b,c). Although EAG responses to each concentration of 2-phenylethanol were not significantly different between virgin and mated females, mated flies exhibited generally higher antenna responses to all concentrations of 2-phenylethanol, except 0.04% (*p* < 0.05; Figure 5a). When the overall EAG responses to the three chemicals were compared between the two females based on the slope values of linear regression, the antenna of mated *D. melanogaster* showed 7.2-, 1.6-, and 32.3-fold higher concentration-dependent EAG responses to 2-phenylethanol, ethanol, and acetic acid, respectively, than the antenna of virgin *D. melanogaster* (Figure 5). 

Based on the comparison of the average EAG response between the two females for each concentration of chemical, mated *D. melanogaster* showed a slightly higher EAG response to 2-phenylethanol (*f* = 4.079, *df* = 34, *p* = 0.061) and a significantly greater EAG response to ethanol (*f* = 10.916, *df* = 34, *p* < 0.001) and acetic acid (*f* = 29.185, *df* = 34, *p* < 0.001) (Figure 6a). This supported the relatively higher chemical sensitivities of the antenna of mated *D. melanogaster*. In addition, a comparison of the ratio of EAG responses between mated and virgin females revealed that mated flies had 1.4-, 2.78-, and 3.67-fold higher responses to the three chemicals than those of the virgin flies (Figure 6b). The ranking of the differences in EAG signals between the two females was in the order of acetic acid, ethanol, and 2-phenylethanol (*f* = 46.586, *df* = 2, *p* < 0.001) (Figure 6b), implying that mated *D. melanogaster* had higher antenna sensitivity to acetic acid than to the other two chemicals, compared with virgin females.

## 4. Discussion

### 4.1. Comparison between D. melanogaster and D. suzukii 

During fruit maturation and fermentation, various volatile chemicals such as 2-phenylethanol, ethanol, and acetic acid are produced [3,6,7]. In particular, colonization by microorganisms, including bacteria and yeast, promotes chemical production; therefore, the concentration and quantity of these chemicals are typically higher in overripe and fermented fruits than in freshly ripening fruits [8,9,10,11]. The habitats of *D. suzukii* and *D. melanogaster* have been considered to be environments with low and high chemical concentrations, respectively. In fact, our recent studies revealed that *D. melanogaster* exhibits a higher tolerance to the volatile chemicals 2-phenylethanol, ethanol, and acetic acid than that of *D. suzukii* [14,15,27].

In addition to the differences in chemical tolerance, the distinct habitats led us to hypothesize that *D. melanogaster* and *D. suzukii* have different preferences for various chemicals. Similarly, Keesey et al. [22] hypothesized that *D. suzukii* is less attracted to fermented food sources, whereas it is more attracted to volatile chemicals emitted during the early ripening stages of fruit development than *D. melanogaster* is. To address this hypothesis, Keesey et al. [22] conducted choice assays and electrophysiology experiments; however, both *D. suzukii* and *D. melanogaster* exhibited similar sensitivities and behavioral preferences for volatiles associated with fruit ripening and fermentation processes. Furthermore, previous studies have shown that fermentation volatiles have been identified to mediate attraction in both *D. suzukii* [5,28] and *D. melanogaster* [3,29]. In contrast, according to previous studies comparing the oviposition preferences of *D. melanogaster* and *D. suzukii* among a broad range of ripening stages of strawberries and puree, *D. suzukii* and *D. melanogaster* consistently preferred to lay eggs in fruits at the early and late stages of maturation, respectively [20,21,30], which cannot explain the similar preferences of the two flies for volatile chemicals [22]. Considering that the quantity and concentration of volatile chemicals increase depending on the fruit fermentation and decay process, the different preferences of the two species for various concentrations of chemicals may explain the distinct host preference of the two flies. The findings of the present study support this hypothesis based on a comparison of Y-tube olfactometer assay results and EAG responses between *D. melanogaster* and *D. suzukii* to various concentrations of 2-phenylethanol, ethanol, and acetic acid (Figure 1, Figure 2 and Figure 3). In particular, the Y-tube attraction assay strongly indicated that *D. melanogaster* was attracted to all three chemicals in a concentration-dependent manner, as indicated by the positive linear regression slopes (Figure 1). In contrast, the slope values of *D. suzukii* for all the chemicals were negative, indicating that *D. suzukii* was attracted to low concentrations of each chemical (Figure 1). The EAG responses further supported the finding that *D. melanogaster* was more sensitive to high concentrations of chemicals than *D. suzukii* was (Figure 2 and Figure 3a). Unlike the Y-tube assay (Figure 1), the linear regression slope values for *D. suzukii* were not negative, whereas the slope values for *D. melanogaster* were >3-fold higher than those for *D. suzukii* in response to 2-phenylethanol and acetic acid, suggesting a trend of greater dose-dependent sensitivity of *D. melanogaster* to these two chemicals than that of *D. suzukii* (Figure 2a,c). However, in the case of ethanol, the slope for *D. suzukii* was slightly higher than that for *D. melanogaster* (Figure 2b). In a previous oviposition preference test using three different concentrations of chemicals, *D. suzukii* exhibited a slightly higher preference for a high concentration of ethanol, whereas *D. melanogaster* showed a strong preference for oviposition at every tested concentration of acetic acid [21]. These differences in preference for the type of chemicals between the two flies were further confirmed via comparison of the average EAG response between *D. melanogaster* and *D. suzukii*; the EAG signal of *D. melanogaster* for acetic acid was 2.21-fold higher than that of *D. suzukii*, whereas those of *D. melanogaster* to 2-phenylethanol and ethanol were only 1.31- and 1.31-fold higher than those of *D. suzukii* (Figure 3b). As discussed in a previous study [21], a high concentration of acetic acid is representative of late fruit fermentation stages, which explains the interaction between *D. melanogaster* and high concentrations of chemicals, particularly acetic acid, in the habitat of fruit decay and fermentation.

The EAG responses of *D. melanogaster* were consistently higher than those of *D. suzukii* at all tested concentrations of all three chemicals (Figure 2 and Figure 3a), indicating a higher sensitivity of *D. melanogaster* to volatile chemicals. Although the EAG responses between the two flies were not directly compared, two previous studies have revealed the respective strong and relatively weak EAG responses of *D. melanogaster* and *D. suzukii* to acetic acid and ethanol [3,28]. The great response of EAG to acetic acid has been discussed in a previous study [28], in which a highly selective olfactory neuron for the detection of acids has been found to be generally related to acid-avoiding behavior in *D. melanogaster* [31], and olfactory inputs mediate positional avoidance for acetic acid-containing food [32]. However, in the present study, according to the EAG (Figure 2) and Y-tube (Figure 1) experiments, olfactory sensing of acetic acid and two other chemicals triggers attraction behavior to the chemicals, rather than avoidance behavior. This explains the chemically rich habitat of *D. melanogaster*.

### 4.2. Comparison between Virgin and Mated D. melanogaster

According to previous studies, female *D. suzukii* have a greater preference for fresh strawberries than for overripe strawberries during oviposition [20], whereas adult female flies are also more attracted to overripe strawberries than to green, white, blush, and red strawberries [22]. These two contrasting results have been discussed: virgin females of *D. suzukii* are initially more attracted to overripe fruit volatiles for feeding and mating; however, mated flies are more attracted to fresh fruit volatiles for oviposition, suggesting that the physiological status of female *D. suzukii* may alter the chemical or host preferences [22,23]. In contrast to *D. suzukii*, *D. melanogaster* females preferentially oviposit on a food product containing acetic acid relative to an acetic acid-free product [33]. Similarly, mated female *D. melanogaster* exhibited a relatively higher preference for high concentrations than for low concentrations of acetic acid and the other two chemicals used in the present study (Figure 4). This supported the fact that *D. melanogaster* has a preference for habitats containing high concentrations of chemicals produced by microorganisms during fruit decay or fermentation. Notably, virgin females exhibited different preferences for the three chemicals: a dose-dependent preference for 2-phenylethanol, a strong inverse preference for ethanol, and an almost moderate preference for acetic acid (Figure 4). Although 2-phenylethanol can be largely produced by fermentation upon yeast colonization [34], it has pleasant fruity and floral odors and is the major constituent in scents of many flowers and fruits in nature [35]. Therefore, 2-phenylethanol is a representative chemical for fresh fruit rather than over-ripened fruit. In contrast, during fruit fermentation, ethanol is produced by alcoholic fermentation [36] and acetic acid is subsequently produced from ethanol by acetic acid bacteria [37]. Considering that 2-phenylethanol, ethanol, and acetic acid are sequentially emitted from fresh fruit at each stage of fermentation, virgin female *D. melanogaster* are attracted to the early stage of alcoholic fermentation rather than to fresh fruits, late fermentation of alcohol, or acetic acid fermentation (Figure 4). After mating at the early stages of fermentation, mated females appeared to be attracted to an environment containing higher amounts of volatiles for oviposition. This is further supported by the EAG response (Figure 5). When the ratios of the EAG signal between mated and virgin females of *D. melanogaster* were compared among the three chemicals (Figure 6b), the antenna signal of mated females to acetic acid was considerably greater than that of virgin females, whereas the disparity of the EAG signal to 2-phenylethanol between virgin and mated flies was significantly lower than that for ethanol and acetic acid, suggesting that mated females are more attracted to more fermented or decaying fruits. Although it is still unclear why virgin and mated *D. melanogaster* have distinct preferences for different chemicals and concentrations, the present study revealed that the physiological status (virgin vs. mated) of females is an important factor in possessing different environmental preferences.

## 5. Conclusions

Taken together, based on the Y-tube olfactometer assay and EAG analysis of *D. melanogaster* and *D. suzukii*, our results show that *D. melanogaster* has a higher preference for high concentrations of chemicals than that of *D. suzukii*, consistently supporting the idea that the habitats of *D. melanogaster* and *D. suzukii* are chemically rich and poor environments, respectively. In addition, mated female *D. melanogaster* preferred higher concentrations of chemicals than the virgin flies. In particular, acetic acid represents a more important chemical for attracting mated females for oviposition than 2-phenylethanol and ethanol. Considering that 2-phenylethanol, ethanol, and acetic acid are sequentially emitted from fresh fruit, alcoholic, and acetic acid fermentation, respectively, the EAG signal distances between *D. melanogaster* and *D. suzukii* and between mated and virgin *D. melanogaster* further suggested that *D. melanogaster* prefers a more fermented environment for oviposition. Based on our current results, the different preference for or repellence to various concentration of chemicals between two flies (*D. melanogaster* and *D. suzukii*; virgin and mated female of *D. melanogaster*) might be suggested to be applied for fly control.

## Figures and Tables

**Figure 1 insects-14-00382-f001:**
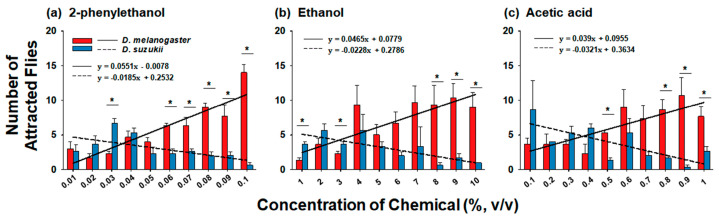
Comparison of the Y-tube choice behavior between *Drosophila melanogaster* and *Drosophila suzukii* to (**a**) 2-phenylethanol, (**b**) ethanol, and (**c**) acetic acid. Twenty flies were introduced into the entrance of the olfactometer, and the number of flies that entered the side arms was counted. At the given concentration of the chemicals, Student’s *t*-test was used to compare chemical preference between the two fly species. Asterisks indicate significant differences (*p* < 0.05). The data are presented as the mean ± standard error.

**Figure 2 insects-14-00382-f002:**
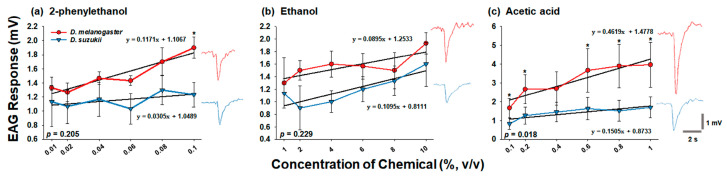
Comparison of EAG responses between *Drosophila melanogaster* and *Drosophila suzukii* to (**a**) 2-phenylethanol, (**b**) ethanol, and (**c**) acetic acid. The EAG response values of the two flies were statistically analyzed using GLM-repeated measures ANOVA. At the given concentration of the chemicals, Student’s *t*-test was used to compare EAG values between the two fly species. Asterisks indicate significant differences (*p* < 0.05). The data are presented as the mean ± standard error. Waveforms of EAG responses of *D. melanogaster* (red) and *D. suzukii* (blue) to (**a**) 0.1% 2-phenylethanol, (**b**) 10% ethanol, and (**c**) 1% acetic acid (**c**). Note that the recovery time of EAG was longer in acetic acid stimulations than that in other stimulations. EAG, electroantennogram.

**Figure 3 insects-14-00382-f003:**
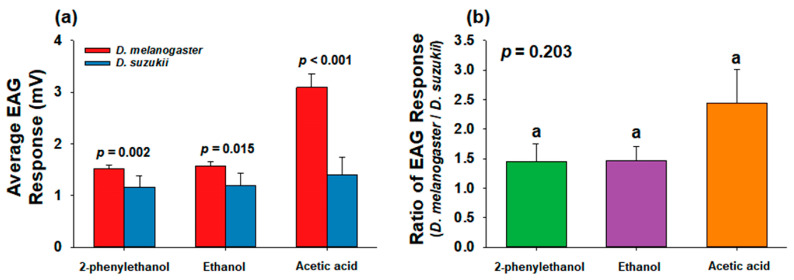
Comparison of (**a**) integrated EAG response values between *Drosophila melanogaster* and *Drosophila suzukii* and (**b**) ratio of EAG response values between two flies (*D. melanogaster*/*D. suzukii*). Student’s *t*-test was used to compare the (**a**) average EAG values between the two fly species (*p* < 0.05) and (**b**) ratios of EAG response value were statistically analyzed using one-way ANOVA. Data are presented as the mean ± standard error. EAG, electroantennogram.

**Figure 4 insects-14-00382-f004:**
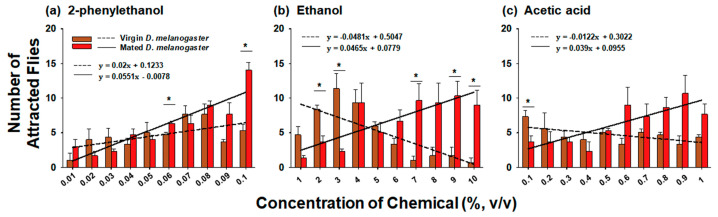
Comparison of the Y-tube choice behavior between virgin and mated female *Drosophila melanogaster* to (**a**) 2-phenylethanol, (**b**) ethanol, and (**c**) acetic acid. Twenty flies were introduced into the entrance of the olfactometer, and the number of flies that entered the side arms was counted. At the given concentration of the chemicals, Student’s *t*-test was used to compare chemical preference between virgin and mated females. Asterisks indicate significant differences (*p* < 0.05). The data are presented as the mean ± standard error.

**Figure 5 insects-14-00382-f005:**
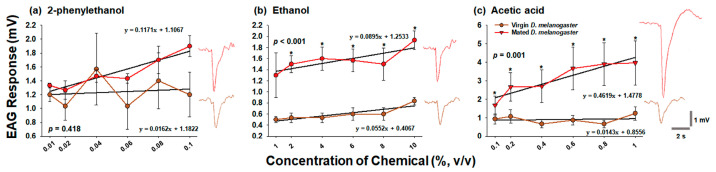
Comparison of EAG responses between virgin and mated female *Drosophila melanogaster* to (**a**) 2-phenylethanol, (**b**) ethanol, and (**c**) acetic acid. The EAG responses values of the two flies were statistically analyzed using GLM-repeated measures ANOVA (*p* < 0.05). At the given concentrations of the chemicals, Student’s *t*-test was used to compare EAG values between virgin and mated females. Asterisks indicate significant differences (*p* < 0.05). The data are presented as the mean ± standard error. Waveforms of EAG responses of virgin (brown) and mated (red) female *D. melanogaster* to (**a**) 0.1% 2-phenylethanol, (**b**) 10% ethanol, and (**c**) 1% acetic acid. Note that the recovery time of EAG was longer in acetic acid stimulations than that in other stimulations. EAG, electroantennogram.

**Figure 6 insects-14-00382-f006:**
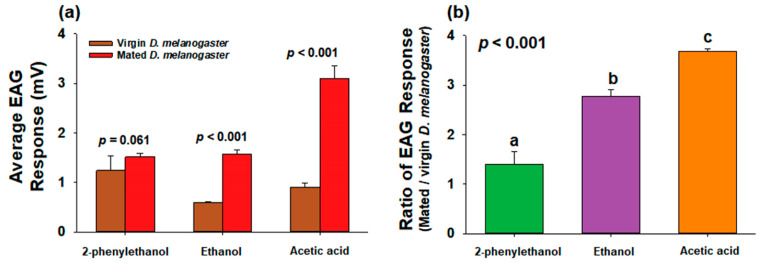
Comparison of (**a**) integrated EAG response values between virgin and mated female *D. melanogaster* and (**b**) ratio of EAG response values between two flies (mated/virgin). Student’s *t*-test was used to compare the (**a**) average EAG values between the two females (*p* < 0.05) and the (**b**) ratios of EAG response values were statistically analyzed using one-way ANOVA. Data are presented as the mean ± standard error. EAG, electroantennogram.

## Data Availability

All of the available data are produced or analysis with this manuscript.

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
