# Peer review of "Comparison of Preference for Chemicals Associated with Fruit Fermentation between Drosophila melanogaster and Drosophila suzukii and between Virgin and Mated D. melanogaster"

_insects, 2023, doi:10.3390/insects14040382_

Round 1

Reviewer 1 Report

The manuscript by Kim and colleagues examines the olfactory preference in D melanogaster and D suzukii for chemicals associated with fruit fermentation. The Y maze behavioral assay and the electroantennogram recording revealed that the D melanogaster flies showed a preference for relatively higher concentrations of 2-phenylethanol, ethanol, and acetic acid than that of D suzukii.

The observations mentioned in the manuscript are interesting. The experiments appear generally well-executed. However, I think the manuscript suffers from several weaknesses that I kindly ask authors to address before I recommend it for publication.

1.    Please mention reproductive status instead of just ‘status’ in line # 137.

2.    The authors mentioned that D Suzukii displayed concentration dependent EAG response to three concentrations (Line # 234). However, in Figure 2, the dose response graph shows that there is no change in EAG amplitude with increased concentrations for 2-phenyl ethanol and acetic acid. The statement does not match with the results. The EAG response appears flat for D Suzukii in those two specific odorants. This could be due to the fact that D Suzukii is less sensitive to those odorants tested. I request authors to modify the statements to match their results. It seems to me that the behavioral response of D Suzukii cannot be predicted from the EAG response. Authors found that flies showed increased preference at lower concentrations (0.03%) of 2-phenyl ethanol when compared with higher concentrations of the same odorant (0.1%) (the slope value was negative), However, the authors failed to see any significant change in EAG amplitude across those concentrations. I recommend the authors include this in the discussion section.

3.    What is the age of the mated female flies? In the methods, authors mentioned that the age of virgin female flies is <1 day old. Were age-matched mated female flies used to compare the EAG response to the virgin female flies? Authors can keep the virgin female flies in a food vial without male flies for 3-5 days, then perform the EAG response with age-matched mated female flies.

4.    No concentration dependent EAG response was observed for ethanol and acetic acid in virgin female flies. It is unclear whether this is due to the virgin female flies being younger than the mated female flies, or the reproductive status of the flies. There is no established determinant relationship.

5.    I request authors to provide the number of flies used for making all the graphs in the legends.

Author Response

Please find attached file for response to reviewer1's comments.

Reviewer 2 Report

Comments and Suggestions for Authors

The manuscript is well written, clearly addressing the preference behaviors of D. melanogaster and D. suzukii. As the eco-friendly management of these two flies rely on the deep understanding of their behaviors, submitted manuscript is good enough to publish. The statistics seemed adequate and clearly showing the principal results and therefore the conclusions seemed robust enough for publishing.

There are only two suggestions for authors:

1.      Statistical analysis section: I’m wondering if the authors have applied normality and homogeneity tests on the data, and if any data transformation was applied before selecting the analysis method? I assume yes, if this the case I recommend mentioning that in the text. If not, I would invite the authors to explain why these tests were not applied.

2.      In the results’ section: I would suggest that f and df values are inserted when possible in the text with p values.

Author Response

Please find attached file for response to reviewer2's comments.

Reviewer 3 Report

The authors have conducted an important study to reveal the difference between two key species of Drosophila and their preference for host volatiles. The experiment selected (Y-tube bioassay) is suitable for this type of work and pairing with EAG analysis, provides solid evidence for observed ecological and behavioral differences between these two species.  I also found the manuscript to be well-written.  There are, however, some issues which need to be addressed before the manuscript is ready for publication. 

1. methods - it is not clear from the methods how many replicates of each pairing were done, nor is it clear why the pairings were chosen... why not test each concentration against a blank?

2. statistical analysis - the design is paired, and using data from a paired design in a repeated measures analysis is not correct.  Repeated measures is a very specific analysis reserved for situations where the SAME individuals are measured before, then after a treatment.  Unless the same 20 flies were used for ALL the experiments, this is not the appropriate analysis. 

Although the results are likely sound and the subsequent discussion and conclusion will likely not change, the statistics need to be redone to match the experimental design.

Detailed comments for the introduction, methods and some of the figures are below.  Until the issues described above are dealt with, there is no point in reviewing the results/discussion/conclusion.

L69 – define ‘evolutionary adapted’. Adapted to respond or not respond?

L73 – suggest elaborating on what the ‘tolerance’ and ‘susceptibility’ mean in terms of behavior.  D. melanogaster would require high concentrations to elicit a positive response, eg. move toward the source while D. suzukii would do what in response to high concentrations?  Be repelled, or simply not be attracted?  Would be good to clarify these terms with regards to the observed behavior

Look up references 14, 15

L113-114 – this seems to be a statement, could be rephrased as an hypothesis, or refer to some previous research, e.g. ‘Previous work [REF] has found high concentrations of volatiles to be a key factor to attract mated females to appropriate oviposition sites.\

L124 – if providing each dimension, the unit is ‘cm’.  ‘Cm3’ would be if you presented only the final volume.  Also, these are commonly called ‘BugDorms’ and this can replace ‘netted cage’ throughout the manuscript.

L126 – ‘eclosed’, not ‘enclosed’

L127 – ‘placed in a netted cage for 3 d and allowed to mate. Virgin females…\

L127 – how many females were in each BugDorm?  How many males?

L137 – was the Y-tube glass?  Need to state this

L145 – how were these solutions provided in the arm of the Y-tube?  If the ‘glass jar’ referred to on L139 was used, need to explain how you created the solutions.  What was the solvent used?  What volume was introduced into the jar and allowed to evaporate?  Need to provide more detail on how these solutions were prepared and how the volatiles were introduced into the Y-tube arms. Method of delivery, time between trials and method of clean up are all critical to ensure no carry over between replicates or treatments.

L145-147 – please clarify why you performed the highest to lowest, then in increments down to the middle as pairings?  What is the justification for doing it like this?

L150 – what comprised an experiment?  Each comparison (as described on L145-147)?  How many replicates (is this what you mean by ‘trial’ on L153?) of each pairing did you do?

L153 – what comprised a trial? Is this your replicate?  And 3 for each pairing?

L161-163 – need a similar description for the Y-tube bioassays and how the odors were prepared and provided into the Y-tube apparatus

L179-181 – I agree that the EAG responses where the same antennae was used for 3 trials can be used in a repeated measures analysis, but I don’t believe the attraction rate trends would meet the assumptions as it was new batches of flies used?  This aspect is not clear from the methods.

L181 – as these are paired responses, I’m pretty sure you cannot do repeated measures across the concentrations as you’ve done here.  Repeated measures need the SAME individuals tested multiple times.  If this is what was done that needs to be stated more clearly in your methods.  As your methods read now, each of your experiments consisted of a pair of concentrations with 20 flies exposed.  Given each experiment lasted 45 minutes and there’s no mention of providing food or water to the flies, it is difficult to believe that the same 20 flies were used for EACH experiment. Additionally, it is a paired design.  Had you done each of these concentrations against a blank control, I could see better justification for presenting your data as you have and conducting a regression (and comparing regression equations across each fly type – mated/virgin, D.melano/D.suzu), but as the data were generated using a paired design, each of your concentrations is naturally going to be connected to one of the other points, e.g. if 20% of the flies went to 0.01% that means there’s only 80% to go to the other side.  All of your percentages are skewed for this reason.

L195 – this is not clear.  Was the choice between 0.01 and 0.1% 2-phenylethanol? Or was the choice between various concentrations and 0.1%.  The statement on L196 would suggest multiple concentrations included in this sentence.

Note: would like to see the percentage of non-responding flies for each experiment.

Figure 1 – y-axis: ‘Attracted flies (%)’

Figure 2 – again, I don’t think you can pull a repeated measures on these data.  You can run separate linear regressions then compare these between the flies, but repeated measures suggest that you used the same flies and took repeated measurements on the same flies and you did not. The only data where repeated measures could be used are for the flies where you recorded their response 3 times for specific concentrations.

Figure 3 – in (a) don’t need the line from one bar to the other, the asterix over the interface of the bars is sufficient to denote significance.  I would also suggest putting the p-value above the bars rather than an asterix as the significance for 2-phenylethanol and ethanol is likely not as significant as the p-value for acetic acid. In (b) – what’s the value of showing the ratio?  The bars in (a) provide enough data, (b) seems repetitive.

Author Response

Please find attached file for response to reviewer3's comments.

Round 2

Reviewer 3 Report

I commend the authors on their revision of the manuscript.  They have acknowledged many of the suggestions and incorporated these into the revised manuscript.  There remain, a few issues which need to be clarified.

1. L70 - need to define 'be tolerant to'.  What does this mean?  they are neither attracted nor repelled from these odors?  

2. L75 - this opens a whole set of questions.  How were these flies 'treated with' 2-phenylethanol?  and why?  do the authors mean that they reared them in an environment where they were exposed to a high concentration of 2-phenylethanol?  were these concentrations similar to what they'd be exposed to in the field?  need to provide more detail on this as it relates directly to 'tolerate' on L70. 

3. L128 - what is the usual sex ratio for these flies?  50:50?

4. L135 - i think you mean 'not receptive' until 24 h after eclosion

5. L151 - I think I understand the set up better although if you applied 2 mL of your solutions 'to each arm' - how was this achieved?  the compounds were in water so these volumes were pools of liquid within each arm?  please clarify.  It is not uncommon for compounds to be dissolved in solvent, then the solvent applied to filter paper and evaporate prior to introduction into the arm and you've done something different here.

6. L156 - would like to see the number of 'no choice' for each experiment reported somewhere in the manuscript.

7. L157 - thank you for clarifying the replication. Please clarify how you cleaned out the chemical solutions from each arm before a new replicate was introduced if the olfactometer was only fully cleaned after 3 uses. How can you be sure that there was no residue from the solution remaining in the arm? Also, were the flies able to reach the solutions? or was there a cage that prevented them from moving that far into the arm?

7. L161 - as concentrations were paired should this read 'replications of each pair of concentrations'?

8. L169 - '..as for the Y-tube tests.'

9. L182 - with the new information on L183, I think this line should read ''Each antennae was exposed to all concentrations from a single chemical starting with the lowest and ending with the highest concentration."  If this is correct, the use of GLM-repeated measures within each species is appropriate.

10. L183 - thank you for clarifying that each replicate was a single antennae.  

11. L203 - suggest rewording "D. melanogaster and D. suzukii exhibited significantly difference preferences to each chemical and concentration range tested using a Y-tube bioassay (Figure 1 and S1)."

12. L208 - similar rewording suggestion "D. melanogaster showed a higher preference for ethanol at concentrations of 8-10% than that of D. suzukii (Figure 1b), while D. suzukii preferred lower concentrations of ethanol."

13. L220 - not sure you need this many significant figures for a slope.  Suggest having only 2, e.g. 0.055, 0.046 and 0.039, and you can delete 'in the conditions of' and replace with 'for' and use 'treatments' (L221)

14. L222 - see comment 13 for significant figures suggestion

15. L300 - 'Figures 4 and S1'

15. L303 - significant figures (see comment 13)

16. L416 - either 'assay' or 'bioassay' - please choose one and be consistent throughout the manuscript

17. final thought - could the authors suggest use of these results as a control?  If D. suzukii is repelled by high concentrations and less ripe fruit is not suitable for D. melanogaster oviposition, could use of high release devices serve to reduce damage from both species?

Author Response

Thank you for your valuable comments. Please see attached file for response to your comments. 
